# Investigating Descending Pain Regulation in Fibromyalgia and the Link to Altered Autonomic Regulation by Means of Functional MRI Data

**DOI:** 10.3390/brainsci14050450

**Published:** 2024-04-30

**Authors:** Shima Hassanpour, Hannan Algitami, Maya Umraw, Jessica Merletti, Brieana Keast, Patrick W. Stroman

**Affiliations:** 1Centre for Neuroscience Studies, Queen’s University, Kingston, ON K7L 3N6, Canada; shima.hassanpour@queensu.ca (S.H.); hannan.algitami@queensu.ca (H.A.); maya.umraw@queensu.ca (M.U.); 18jsm10@queensu.ca (J.M.); 18bkk1@queensu.ca (B.K.); 2Department of Biomedical and Molecular Sciences, Queen’s University, Kingston, ON K7L 3N6, Canada; 3Department of Physics, Queen’s University, Kingston, ON K7L 3N6, Canada

**Keywords:** fibromyalgia, functional magnetic resonance imaging, neural signaling, pain, connectivity, spinal cord, brainstem

## Abstract

Fibromyalgia syndrome (FM) is a chronic pain condition that affects a significant portion of the population; yet, this condition is still poorly understood. Prior research has suggested that individuals with FM display a heightened sensitivity to pain and signs of autonomic dysfunction. Recent advances in functional MRI analysis methods to model blood-oxygenation-level-dependent (BOLD) responses across networks of regions, and structural and physiological modeling (SAPM) have shown the potential to provide more detailed information about altered neural activity than was previously possible. Therefore, this study aimed to apply novel analysis methods to investigate altered neural processes underlying pain sensitivity in FM in functional magnetic resonance imaging (fMRI) data from the brainstem and spinal cord. Prior fMRI studies have shown evidence of functional differences in fibromyalgia (FM) within brain regions associated with pain’s motivational aspects, as well as differences in neural activity related to pain regulation, arousal, and autonomic homeostatic regulation within the brainstem and spinal cord regions. We, therefore, hypothesized that nociceptive processing is altered in FM compared to healthy controls (HCs) in the brainstem and spinal cord areas linked to autonomic function and descending pain regulation, including the parabrachial nuclei (PBN) and nucleus tractus solitarius (NTS). We expected that new details of this altered neural signaling would be revealed with SAPM. The results provide new evidence of altered neural signaling in FM related to arousal and autonomic homeostatic regulation. This further advances our understanding of the altered neural processing that occurs in women with FM.

## 1. Introduction

Fibromyalgia (FM) is a chronic pain condition affecting 2% to 4% of the population [1] and is characterized by musculoskeletal pain, fatigue, cognitive disturbance, hyperalgesia, and allodynia, and affects predominantly women [2]. This condition remains poorly understood, but a number of possible mechanisms have been proposed involving altered central pain processing, including the central sensitization or alterations in descending pain regulation, endocrine functions, stress responses, or cognitive or emotional processes [3,4]. Functional MRI studies have provided new insights into altered neural processes in patients with FM. Based on studies by Staud et al. [3], which demonstrated an increased temporal summation of second pain (TSSP) in FM; many of the subsequent functional magnetic resonance imaging (fMRI) studies employed calibrated thermal stimuli designed to evoke temporal summation. However, some of the first brain fMRI studies of responses to noxious stimuli showed similar blood-oxygenation-level-dependent (BOLD) responses when the stimuli were calibrated to produce a similar pain in patients with FM compared to healthy controls (HCs) [2,5]. Functional MRI studies involving the brainstem and spinal cord identified regions in the periaqueductal gray (PAG), nucleus tractus solitarius (NTS), locus coeruleus (LC), and spinal cord dorsal horn, which appeared to have lower BOLD responses in FM compared to HC [2]. A later study demonstrated the altered neural signaling and co-ordination between regions (i.e., connectivity) in FM patients both while anticipating and experiencing noxious stimuli [6]. The results indicated that women with FM may have altered nociceptive processing in systems involved with pain regulation, arousal, and autonomic homeostatic regulation. The results of the previous studies in our lab have demonstrated the continuous nature of pain regulation, such as BOLD responses in the brainstem and spinal cord regions to participants being informed of a pending stimulus and experiencing it, and, also, changes in BOLD fluctuations after the stimulus has passed [7,8]. We, therefore, developed a novel connectivity analysis method that incorporates information about BOLD responses and prior information about anatomy and physiological responses in order to extract more information from fMRI data. This method is called structural and physiological modelling (SAPM) [5,9]. The purpose of the present study was to apply SAPM to fMRI data from a prior study in order to better investigate the altered nociceptive processing in the brainstem and spinal cord than was previously possible and to identify how it is altered in FM compared to HC individuals. We hypothesized that SAPM would demonstrate altered neural signaling in FM compared to healthy controls (HCs) in areas of the brainstem and spinal cord that are involved in autonomic regulation and descending pain regulation, including the parabrachial nuclei (PBN) and nucleus tractus solitarius (NTS).

## 2. Materials and Methods

For this study, we used data collected previously in our lab from 15 females with FM and 15 females without any chronic pain conditions for comparison, and which are referred to as healthy controls (HCs) [7,10]. Women were chosen for this study because FM is much more prevalent in women than in men. The participants with FM had been previously diagnosed by a physician as having FM, and they fulfilled the 1990 and 2016 FM criteria [11,12] as confirmed by pressure-point testing and questionnaires. Participants in the healthy control group were free of any neurological disorders or major illnesses and were not taking any centrally acting medications. All study methods were reviewed and approved by our institutional research ethics board, and participants provided fully informed written consent before participating.

### 2.1. Participants

FM participants had been identified previously according to the 1990 and 2016 diagnostic criteria for FM criteria, and all participants (FM and HC) met the prerequisites for magnetic resonance imaging (MRI) without contraindications. HC participants were free of medications affecting the central nervous system, and FM participants using medication to treat their condition were allowed to maintain their regimen. Individuals on centrally acting pain medication were excluded from the study.

### 2.2. Questionnaires and Training Session

Participating in the study involved one visit to our institution’s MRI facility, and the visit consisted of a training session followed by the MRI session. Before beginning the training session, participants were assessed with questionnaires that included demographic information, mental health, pain symptoms, and autonomic functioning. The 2016 Fibromyalgia Survey Questionnaire (FSQ) [11] was administered to adhere to the latest fibromyalgia diagnostic criteria. A tender point test based on the ACR FM criteria [11,13] was also applied using an algometer and focused on 12 points above the waist. Participant characteristics were quantified with the Beck Depression Inventory (BDI) [14], the State-Trait Anxiety Inventory [15] (STAI), the Social Desirability Scale (SDS) [16], and the Pain Catastrophizing Scale (PCS) [17]. Autonomic health was also assessed with the Composite Autonomic Symptom Score 31 (COMPASS-31) [18], covering six autonomic symptom severity areas. Pain and related symptoms were evaluated using the Revised Fibromyalgia Impact Questionnaire (FIQR) [19] and the Short-Form McGill Pain Questionnaire-2 (SF-MPQ-2) [20]. The SF-MPQ-2 included four subsections addressing diverse pain quality aspects, with group averages computed and compared between the FM and HC participants. The psychological data assessment reports were excluded from the present research, as the study focused on connectivity analysis. All of the questionnaire results have been reported previously [7,8].

The training session involved familiarizing the participant with the study paradigm, and exposure to a sham MRI in the training session room in order to reduce participant anxiety. The participants were trained how to use a numerical pain intensity scale (NPS) with a 0 to 100 range [21] for reporting their pain experiences. The stimulus consisted of a calibrated noxious heat stimulus, as described below.

### 2.3. Thermal Stimulation

This study used a custom-made MRI-compatible robotic contact-heat thermal stimulator (RTS-2) to administer the heat stimulus. The RTS-2 was made to facilitate comparisons with recent pain research in the spinal cord and to consider the known association between fibromyalgia and heightened sensitivity to heat pain. The stimulator consists of a plexiglass case housing a heated aluminum thermode, which advances to contact the participant’s skin, and then retracts back into the case. The movement and temperature of the thermode are controlled by custom software developed in MATLAB (MathWorks Inc., Natick, MA, USA). Participants were told to put the “heel” of their right hand on the thermode. This location activates the right dorsal horn at the sixth cervical (C6) spinal cord segment and allows us to compare our results with previous research [22,23,24]. In the training session, a calibration procedure was carried out to determine the required temperature to induce moderate pain in each participant. This procedure also allows the participant to practice using the pain-rating scale. Each period of stimulation consisted of 10 heat contacts to the hand, which lasted 1.5 s each, with onsets occurring every 3 s over a 30-s period. The paradigm selected for studying FM is based on the understanding that FM involves central sensitization and exhibit increased temporal summation of second pain [21,25,26,27]. Each participant received the same heat stimulation temperatures of 46, 50, 44, and 48 °C (one temperature tested in each run), and they were instructed to report their pain rating as they experienced each contact. Participants were not informed of the specific temperatures used during the tests to avoid biasing their responses. Participants also practiced the experimental protocol in a sham MRI scanner following the training session. During practice and actual fMRI runs, they were instructed to mentally rate their pain for each contact and remember their ratings for the first and last contacts. This was to avoid overt speech which could introduce head motion. They were then asked to verbally report their ratings at the each of each run.

### 2.4. FMRI Stimulation Paradigm

The imaging session consisted of 10 fMRI runs, each lasting 270 s (Figure 1). These runs were divided into five “Pain” runs during which participants were subjected to the noxious heat stimulus, interspersed in a random sequence with five “No-Pain” runs, where participants did not experience the stimulation. Within each run, at the 1-min mark, participants were informed whether they would receive the stimulus in that run. If it was a “Pain” run, participants were alerted at the 1-min mark that the stimulation would commence in 1 min. During the stimulation phase, they encountered 10 heat contacts over 30 s, with temperatures set at the calibrated level, as in the sham training session. The NPS was displayed during this period, and participants were instructed to mentally assess and rate each contact. After the stimulation phase, imaging continued for another 2 min. Following each “Pain” run, participants were asked to verbally report their ratings for the first and last contact through a two-way communication system, and they were informed that another run would commence shortly. The “No-Pain” imaging process had the same duration, but participants were told they would not receive the stimulus. This paradigm has been used in previous pain studies [22,23,24,28].

### 2.5. FMRI Data Acquisition

This study utilized data from a broader research program encompassing brainstem and spinal cord imaging sessions and brain imaging sessions. However, this specific study focuses solely on the brainstem and spinal cord imaging data. FMRI scans were conducted using a Siemens 3-Tesla MRI system. Functional images were obtained using a half-Fourier single-shot fast spin-echo (HASTE) sequence with BOLD contrast, encompassing the entire brainstem and cervical spinal cord. This method has been demonstrated to provide optimal image quality and BOLD sensitivity for the brainstem and spinal cord and has been validated in numerous previous studies [24]. The 3D volume was imaged with 9 contiguous sagittal slices, each 2 mm thick, covering a field of view of 28 × 21 cm and an in-plane resolution of 1.5 × 1.5 mm. The imaging parameters included an echo time (TE) of 76 ms and a repetition time (TR) of 6.75 s/volume, which was selected for optimal T2-weighted BOLD sensitivity [29]. Each imaging run comprised 40 volumes, equivalent to a 4.5-min duration. In total, each participant underwent 10 runs, divided into 5 “Pain” runs and 5 “No-Pain” runs, resulting in 200 volumes for each condition per participant [23].

### 2.6. Analysis Method

#### 2.6.1. FMRI Data

The fMRI data were analyzed using a custom Python software package called “Pantheon”, which was developed in our lab and is freely available on GitHub (https://github.com/stromanp/pantheon-fMRI, accessed on 29 April 2024). This software includes specialized methods for analyzing fMRI data from the brainstem and spinal cord. The imaging data were first transformed from DICOM to NIFTI format. Subsequent pre-processing steps included motion correction (co-registration), slice timing correction, interpolation to 1 mm cubic voxels and spatial normalization, and modeling and removal of physiological noise and motion-related signal variations, as described previously [5,9].

#### 2.6.2. Anatomical Regions and the Network Model

Anatomical regions used for SAPM were identified using the region maps compiled from multiple sources as described previously [5,9]. Each region was then divided into five sub-regions of near-equal volumes by means of k-means clustering. The choice of five sub-regions is based on previous studies to allow for functional variations within regions [24,28,30,31]. The regions included the right dorsal horn of the sixth cervical spinal cord segment (C6RD), the dorsal reticular nucleus of the medulla (DRt), the hypothalamus, locus coeruleus (LC), nucleus gigantocellularis (NGc), nucleus raphe magnus (NRM), nucleus tractus solitarius (NTS), the periaqueductal gray (PAG) region, parabrachial nuclei (PBN, medial and lateral division), and the medial thalamus. Plausible anatomical connections between these regions were based on prior studies as well, and the detailed descriptions by Millan [32], and a network model was created as shown in Figure 2. This model has been previously described and validated [5,9].

#### 2.6.3. Structural and Physiological Modeling (SAPM)

Structural and physiological modelling (SAPM) uses information related to anatomy, neurophysiology, and blood-oxygenation-level-dependent (BOLD) MRI signal variations to model the neural signaling underlying observed BOLD responses across interconnected networks of regions. This method has been described in detail and validated previously [5,9]. The method requires a predefined network model which specifies the anatomical regions involved, and plausible anatomical connections between them. The model network includes latent or intrinsic inputs from outside of the network, and these latent inputs drive the signal variations across all regions of the network. The output signaling from each region is modeled as inputs to other regions, and scaling factors, termed DB values, relate how an input to a region is scaled to contribute to the output signaling from the region. Positive DB values correspond with excitatory input (more input signaling results in more output signaling) whereas negative DB values reflect inhibitory input (more input signaling results in less output signaling). The output from one region can be modeled as inputs to multiple regions, and different amounts of contributions are scaled by “D” values. The total input to each region is summed to model the observed BOLD responses. Thus, the SAPM method models both input and output signaling to/from each region as signals that are equivalent to BOLD responses and relate to variations in metabolic demand of the regions. By fitting observed BOLD responses to the network model, it is possible to determine the connectivity DB values, D values, and time-courses of latent inputs. The results provide models of the input and output signaling from each region. SAPM was applied to data from one participant at a time in order to identify individual variability across the study groups. Significant DB values were identified as those with group averages that are significantly different than zero, or that are correlated with pain responses. Analyses of variance (ANCOVAs) were also applied to identify any relationships between connectivity values and pain responses.

## 3. Results

### 3.1. Demographical Information

Demographic information for the HC and FM groups are repeated from the previous study in our lab [8,23] presented in Table 1. The normalized pain score was calculated by dividing each participant’s average pain rating by the average stimulus temperature needed to elicit that rating. A higher number indicates a higher pain sensitivity. FM participants are shown to have a higher average age and normalized pain score and show higher initial pain scores (ratings for the first heat contact) than healthy controls. These findings are consistent with previous studies showing that individuals with FM have a higher pain sensitivity compared to the healthy controls. Furthermore, the FM group exhibited higher pain responses to the first and last thermode contact (i.e., “First and Last” pain scores), before and after the effects of temporal summation with repeated contacts.

### 3.2. Structural and Physiological Modeling Results

The SAPM results are shown as connectivity plots in Figure 3, demonstrating the differences in signaling between the groups (FM and HC) in two conditions (Pain and No-Pain). Gray ovals represent the anatomical sub-regions, each labelled with an abbreviated name and a sub-region number. The findings are presented for one distinct set of sub-region combinations. The arrows in the diagrams indicate the direction of signaling, and solid lines signify excitatory effects, and dashed lines signify inhibitory effects. Significant connections in the HC and FM groups, in both the “Pain” and “No-Pain” conditions, are shown in different colors. Significance was based on a t-test comparing group average connectivity (DB) values to reference values from tests with null data, at a corrected *p* < 0.05 (*p* < 0.00156 uncorrected, based on a Bonferroni correction for 32 connections, not counting latent inputs). The corresponding DB values are also listed in Table 2.

The significant connections (Table 2) involved primarily the thalamus, hypothalamus, LC, and PBN. The PBN→thalamus connection was most common across the four groups/conditions. The LC→hypothalamus connection had a higher significance in the FM group for both conditions, but the connectivity strength was not notably different. The connectivity values appear to be more consistent within the FM group for this connection. In contrast, the LC→thalamus connection is stronger and more significant in the HC group, in both conditions. The hypothalamus→LC connection appears to be consistently stronger in the No-Pain condition in both FM and HC groups, but the values are higher in the FM group.

### 3.3. Correlations between Connectivity Values and Pain Ratings

Regression analyses were applied to identify significant relationships between connectivity values and pain ratings within each of the study groups/conditions. Only one connection was found to be significant at a Bonferroni-corrected *p* < 0.05. The pain rating to the first contact was found to be correlated with DB values in the FM group in the Pain condition. The results are plotted in Figure 4, along with the corresponding results for the HC and FM Pain condition. The corresponding anatomical regions are shown in Figure 5.

### 3.4. ANCOVA (Analyses of Covariance) Results

The relationships between study groups/conditions and pain ratings were further investigated by means of analyses of covariance (ANCOVAs). Table 3 lists the results for both the FM and HC groups. There are significant differences between the FM and HC connectivity values, with the C6RD→thalamus connection having higher connectivity values in FM than in HC. The corresponding anatomical regions are shown in Figure 6.

Based on the prevalence of the LC region in the significant connectivity results, this region was selected for demonstrating the time-course responses. Figure 7 shows the input and output signaling to/from the LC in the FM and HC groups, for the Pain condition. The left side of each set of plots shows the modelled output signaling from the PAG and PBN and a latent input, which explains the total input signaling to the LC. The upper plot on the right side of each set of plots shows the observed BOLD responses (averaged across runs) for each group in red, with the modelled input signaling plotted in blue. The lower plot shows the modelled output signaling from the LC in blue.

## 4. Discussion

The results of the present study add new information about the altered neural signaling associated with fibromyalgia. The previous analyses that we reported employed structural equation modeling with the same fMRI data from participants with fibromyalgia, and healthy control participants for comparison [7,8,23]. Those results demonstrated the differences in coordinated signaling between regions including the LC, hypothalamus, PAG, and PBN. Women with FM were concluded to have altered pain regulation that appears to be linked to altered neural signaling related to arousal, and autonomic homeostatic regulation. In the present study, we applied our new analysis method, SAPM, to obtain more detailed information and further investigate how neural signaling involved with nociceptive processing is altered in FM. 

The results summarized in Figure 3 show more significant connections to/from the hypothalamus with the LC and PBN in FM than in HC, whereas, in HC participants, there were more connections that were significant to/from the thalamus with the LC and PBN. The results, therefore, indicate that these regions (LC, PBN, hypothalamus, and thalamus) play important roles in how neural signaling is altered in people with FM, consistent with the previous findings. The connectivity values listed in Table 2 provided greater detail, such as the fact that the PBN→thalamus connection involves predominantly excitatory signaling that is weaker in FM compared to HC. In contrast, the LC→hypothalamus connection is observed to be stronger in FM than HC. It is also notable that the average connectivity value for PBN→thalamus is lower in the Pain condition than the No-Pain condition in the HC group, and the values are lower in both conditions in the FM group. Similarly, the values for the LC→hypothalamus connection are higher in the Pain condition than the No-Pain condition in the HC group, and the values are higher in the FM group in both conditions. These values may reflect the fact that participants in the FM group experience constant pain, including in the No-Pain condition of this study. The differences in connectivity values for these particular connections may, therefore, be caused by chronic pain in FM. 

Other connections that were found to be significant demonstrate different features. The PBN→hypothalamus connection has higher values in the No-Pain condition than the Pain condition, in both FM and HC groups. The hypothalamus→LC connection has consistently higher values in the FM group than the HC group in both conditions, whereas the LC→thalamus and LC→PBN connections have lower values in the FM group compared to the HC group in both conditions. The LC→DRt connection was the only connection that was significant at the group level that has negative values (indicating predominantly inhibitory signaling) and the pattern of differences between groups/conditions is unclear. The LC→thalamus connection was the only one that had values that were significantly correlated with pain ratings across any of the groups/conditions. This connection was observed to have higher values in FM participants with lower initial pain ratings and decreasing values with higher pain ratings. This pattern indicates that LC signaling to the thalamus is excitatory and contributes to reducing pain sensitivity. The results of ANCOVA analyses shows the interaction effects of the groups also indicate differences in the C6RD→thalamus and PAG→NTS connections between groups, but these differences did not reach significance after correcting for multiple comparisons. Nonetheless, these results suggest that there may be differences as well between the FM and HC groups in descending pain regulation pathways, consistent with previous studies [22,24]. 

The results of the SAPM analysis also reveal which anatomical sub-regions within each region have BOLD responses that best fit the model network, and they provide models of the input and output signaling from each region. The BOLD signaling examples in Figure 7 show input and output signaling for the locus coeruleus (LC). BOLD signal variations in the LC are demonstrated to occur after participants were informed of the study condition (Pain or No-Pain) and in response to the stimulus. Differences in input signaling to the LC from the PAG and PBN are also shown between the HC and FM groups at the times when the participants were informed of the study condition, just prior to stimulation, and during stimulation. The modeled latent input also shows variations during these time periods and represents signaling originating from outside of the modeled network, likely cortical regions providing input to the LC.

The results obtained with the SAPM analysis consistently suggest that the LC, hypothalamus, PBN, and thalamus play important roles in how neural signaling is altered in FM, and that these changes are related to altered nociceptive processing. The PBN plays a role in autonomic regulation, and receiving, processing, and relaying nociceptive signals [24]. The LC contributes to regulating pain and stress-related conditions, such as major depressive disorders and anxiety [33]. Furthermore, the LC is part of the norepinephrine system and is involved in resilience, activating the fight-or-flight response, fear learning, autonomic responses, and pain modulation [34]. The thalamus receives input from cortical regions as well as from multiple ascending pain pathways. This region actively processes nociceptive information before transmitting it to various cortical regions [34,35]. The hypothalamus is crucial in regulating autonomic functions, serving as one of the main regions for descending pain modulation. The DRt is among the regions transmitting the pain modulation through direct projections to and from the spinal cord [36]. The results of the present study are, thus, consistent with previous conclusions that FM may involve a convergence of systems that co-ordinate descending pain regulation and homeostatic autonomic regulation. The involvement of the LC, hypothalamus, and thalamus is consistent with previous studies suggesting that FM may result from an altered state or imbalance in endocrine functions, possibly involving a thalamocortical loop, the hypothalamus–pituitary–adrenal (HPA) axis, or other endocrine systems [37,38]. However, it is not clear which aspects of altered signaling between regions in the FM group may be the result of chronic pain, and which may reveal a possible mechanism of how pain sensitivity is altered in FM. 

The results obtained with SAPM are consistent with our prior studies of FM, and they provide further support for the conclusion that FM involves altered autonomic regulation. Martucci et al. carried out a resting-state fMRI study in the cervical spinal cord and the findings support the observation that neural signaling is altered in the spinal cord in people with FM [39]. A theoretical model of altered neural signaling in FM has recently been described by Demori et al. [38] involving feedback between regions of the thalamus and somatosensory cortex. The model proposes a mechanism for an altered stable state of neural signaling in FM that may be influenced, or produced, by an altered immunoendocrine function. This model may be consistent with the observation of altered signaling between the LC and thalamus in the present study. Other possible mechanisms that have been proposed as underlying mechanisms in FM include the dysregulation of norepinephrine [40]. diminished serotonin transporter function [41], hyperactivity of the hypothalamic–pituitary–adrenal (HPA) axis, or a combination of hormonal systems.

The results of the present study are limited by the relatively small sample size. The data used for this analysis were collected previously and data collection was interrupted by the COVID-19 pandemic. These data were used for the present analysis in order to investigate how analyses using our novel SAPM method could provide more detailed information from fMRI data than was previously possible. The results demonstrate considerable detail about the connectivity within/between the brainstem and spinal cord regions in HC and FM participants, and how neural signaling is altered in FM. Future studies will expand on these findings with larger datasets and by incorporating methods to further investigate variations in autonomic function that correspond with the noxious stimulation paradigm.

## 5. Conclusions

Significant differences in connectivity values between the FM and HC groups were identified using SAPM at the level of the brainstem and spinal cord within regions involved in arousal and autonomic homeostatic regulation. The novel SAPM analysis method we employed revealed information about coordinated neural signaling including excitatory and inhibitory signaling that is not provided by previous methods. Connectivity values indicate that, in FM, there is the altered co-ordination of signaling across regions that are involved in pain processing and autonomic regulation. The results identified differences in connectivity primarily between the locus coeruleus (LC), hypothalamus, thalamus, and periaqueductal gray (PAG). These results provide additional evidence for how neural signaling is altered in FM. Future studies are required to further explore the relationships between autonomic function and pain perception in FM patients in greater detail.

## Figures and Tables

**Figure 1 brainsci-14-00450-f001:**
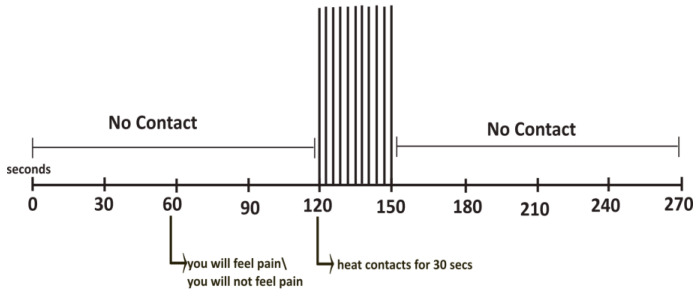
The stimulation paradigm is used for the “Pain” and “No-Pain” conditions. The period between informing participants about the heat stimulus and starting the heat contacts refers to the expectation period.

**Figure 2 brainsci-14-00450-f002:**
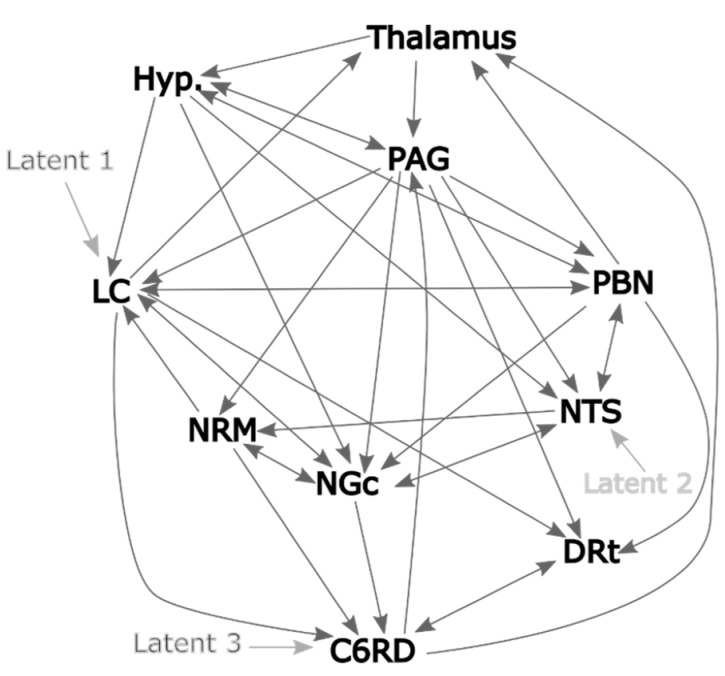
The network model used for the SAPM analysis, with dark grey lines representing the possible network connections and the light grey lines representing the latent inputs. The network model shows 35 connections with 3 latent inputs. The regions included the right dorsal horn of the sixth cervical spinal cord segment (C6RD), the dorsal reticular nucleus of the medulla (DRt), the hypothalamus, locus coeruleus (LC), nucleus gigantocellularis (NGc), nucleus raphe magnus (NRM), nucleus tractus solitarius (NTS), the periaqueductal grey (PAG) region, parabrachial nuclei (PBN, medial and lateral division), and medial thalamus.

**Figure 3 brainsci-14-00450-f003:**
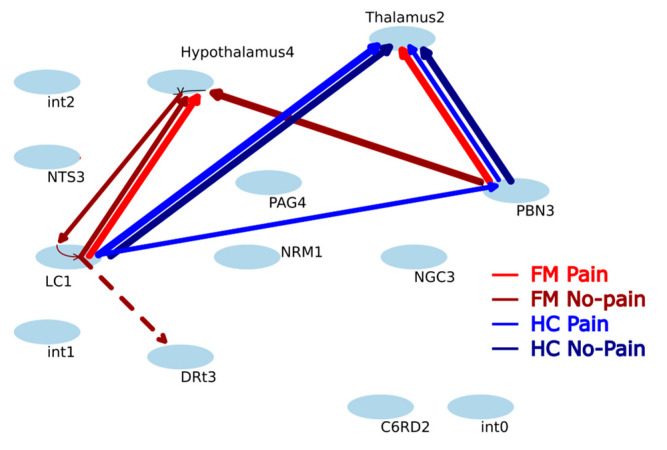
Significant connections identified with SAPM analysis for the four groups/conditions. Ovals indicate anatomical regions; and lines indicate significant positive (solid lines) or negative (dashed lines) connectivity between regions computed with structural and physiological modeling (SAPM). Latent inputs are denoted as “int,” referring to “intrinsic.” Colors represent the different study groups and conditions as indicated in the legend. FM refers to Fibromyalgia, and HC refers to healthy controls.

**Figure 4 brainsci-14-00450-f004:**
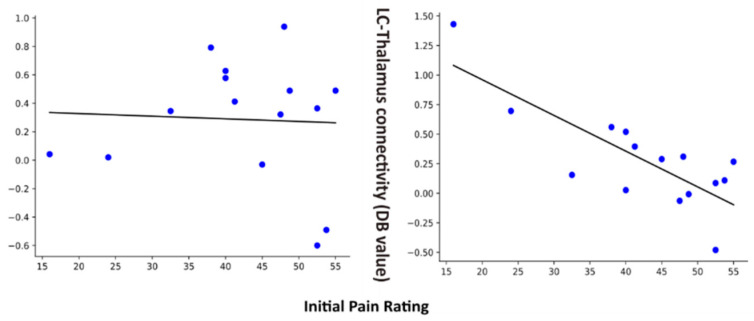
Plots of connectivity values (DB values) versus pain ratings for the selected region (LC to Thalamus) in FM pain and HC pain condition.

**Figure 5 brainsci-14-00450-f005:**
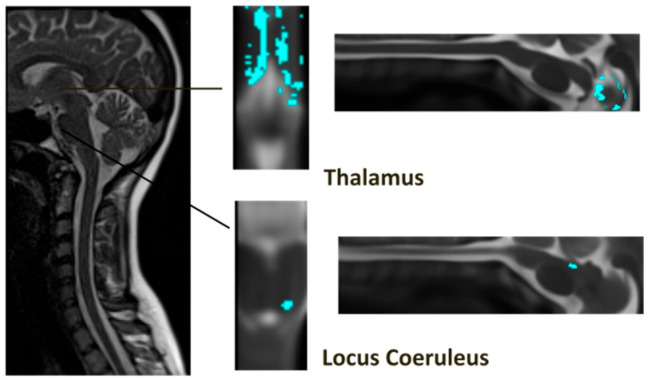
Visual display of LC to thalamus connectivity. The left side panel shows an example of the original MRI data, and the two examples are shown with one sagittal and axial view for each sub-region in spatially normalized format, the blue color indicates the anatomical sub-regions.

**Figure 6 brainsci-14-00450-f006:**
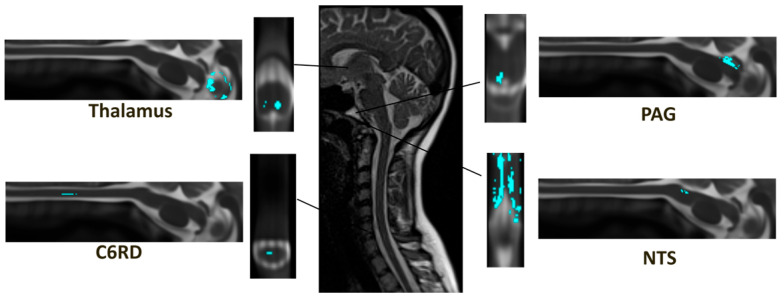
Anatomical regions involved with connections with significant ANCOVA results. The middle panel shows an example of the original MRI data, and the four examples are shown with one sagittal and axial view for each sub-region in spatially normalized format, the blue color indicates the anatomical sub-regions. The connections that are represented include the C6RD→thalamus and PAG→NTS.

**Figure 7 brainsci-14-00450-f007:**
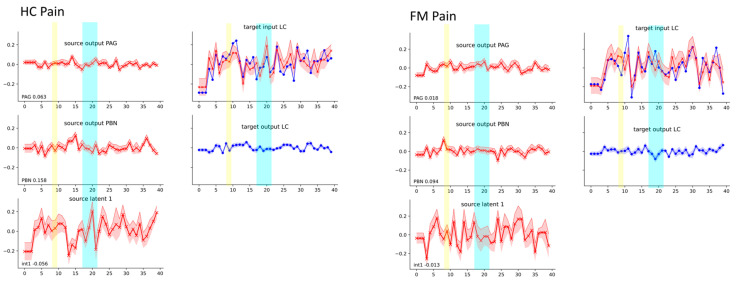
Details of the input and output signaling to the LC in FM and HC participants. The yellow bar indicates when participants were informed that a noxious stimulus would be applied one minute later, and the blue bar indicates the period when the stimulus was applied.

**Table 1 brainsci-14-00450-t001:** Demographic information of healthy (HC) and fibromyalgia (FM) groups. Mean scores, plus or minus the standard deviation, are presented. The normalized pain score was determined by dividing each participant’s average pain rating by the average stimulus temperature required to elicit that rating. A higher value signifies greater pain sensitivity.

Demographic Measures	HC (Mean ± SD)	FM (Mean ± SD)
Age	39.2 ± 10.3	46.7 ± 13.5
BMI	27.6 ± 3.8	25.8 ± 5.1
Normalized Pain Score	0.72 ± 0.2	1.01 ± 0.2
First Pain Score	24.6 ± 14.4	41.6 ± 14.2
Last Pain Score	36.8 ± 14.2	46.1 ± 14.9

**Table 2 brainsci-14-00450-t002:** Connectivity values (DB) determined using data from FM and HC groups, in the Pain and No-Pain conditions by means of SAPM analysis. Values in boldface are significant, and corresponding values that did not reach significance in other conditions are listed for comparison. The *p*-value corrected for multiple comparisons is 0.00156, and the corresponding T-threshold is 3.563.

	Fibromyalgia				Healthy Controls			
	Pain		No-Pain		Pain		No-Pain	
Connections	DB	T	DB	T	DB	T	DB	T
PBN→Thal	**0.166 ± 0.040**	**4.38**	0.185 ± 0.087	2.21	**0.282 ± 0.078**	**3.70**	**0.305 ± 0.078**	**4.02**
LC→Hypo	**0.319 ± 0.107**	**3.58**	**0.436 ± 0.137**	**3.66**	0.376 ± 0.137	3.22	0.255 ± 0.150	2.13
PBN→Hypo	0.165 ± 0.063	2.90	**0.254 ± 0.048**	**5.69**	0.103 ± 0.050	2.40	0.266 ± 0.089	3.20
Hypo→LC	0.227 ± 0.110	2.81	**0.311 ± 0.110**	**3.58**	0.088 ± 0.161	1.05	0.128 ± 0.172	1.22
LC→Thal	0.286 ± 0.107	2.74	0.286 ± 0.108	2.72	**0.528 ± 0.097**	**5.54**	**0.379 ± 0.098**	**3.93**
LC→DRt	−0.074 ± 0.127	−0.25	**−0.438 ± 0.103**	**−3.82**	−0.326 ± 0.119	−2.38	−0.268 ± 0.125	−1.81
LC→PBN	0.119 ± 0.113	1.29	0.324 ± 0.160	2.19	**0.439 ± 0.123**	**3.78**	0.274 ± 0.105	2.86

**Table 3 brainsci-14-00450-t003:** ANCOVA results comparing both groups (FM and HC) connectivity values in the pain condition and their relationship with pain rating at Bonferroni *p* < 0.05.

Connection	Effect	FM (Connectivity Values, DB)	HC (Connectivity Values, DB)	*p*-Value	P-Threshold
C6RD-Thalamus	Interaction	−0.113 ± 0.200	−0.012 ± 0.147	0.0082	0.05
PAG-NTS	Interaction	0.084 ± 0.094	0.120 ± 0.086	0.0234	0.05

## Data Availability

The original data presented in the study are available upon request from the principal investigator due to ethical reasons. Pantheon’s analysis software is freely available on GitHub at https://github.com/stromanp/pantheon-fMRI.

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
