# Peer review of "Investigating Descending Pain Regulation in Fibromyalgia and the Link to Altered Autonomic Regulation by Means of Functional MRI Data"

_brainsci, 2024, doi:10.3390/brainsci14050450_

Round 1

Reviewer 1 Report

Comments and Suggestions for Authors

The authors provide an interesting analysis on pain regulation in fibromyalgia linking to altered autonomic regulation using functional MRI. The manuscript is well structured and well written, however I have some remarks that need to be clarified:

Within the introduction and also later on in the discussion it is stated that women with FM have an altered descending pain modulation compared to healthy controls. However in the material section in remains unclear if only women are included in the study. Please clarify.

Are there any results on FM in men regarding this? Or in general something published on the neuronal network of pain modulation independent of FM?

fMRI acquisition within the brain stem provides some challenges due to bony structures and so on. The authors state a specific protocol is used. Please provide the readers with some information on this issue. The image resolution seams suitable but small. Are there any issues regarding the SNR?

How were the HC chosen? Was any matching procedure applied? The age seems lower in the HC group, are there any effects of age reported already regarding the network?

Please further clarify the novelty of the research and what questions arise of the results that will be tackled by further research.

Minor remarks:

Figure 6 should be Figure 7? (line 296 + line 357)

Please introduce all abbreviations at first appearance (e.g. MRI/BOLD , line 16, fMRI, line 43)

Minor spell checking required

Comments on the Quality of English Language

minor spell checking required

Reviewer 2 Report

Comments and Suggestions for Authors

The relevance of the topic of this study is due to the increase in the number of patients with this pathological condition. The results of this study add new information about altered neural signaling associated with fibromyalgia and central sensitization.

Clarify the method which you used for sample size determination

How confident are you that the changes found on fMRA are due to fibromyalgia and not other neuropathic pain associated with central sensitization?

According to the American Pain Society, there are nine possible sites of fibromyalgia pain. It was no better to choose one of these pain points instead of the heel of the hand. Can you explain please

Where is the conclusion?

References are not formatted by journal rules

Specify abbreviations in detail in Figure 1

Correct in table 1 mean ± s.d to mean ± SD

Clarify the significance of differences between the demographical characteristics of the study groups before treatment.

Please in Table 2 clarify the significance difference between groups based on the P value.

In the discussion, add additional information about previous researches on this topic and compare these results with yours.

Round 2

Reviewer 2 Report

Comments and Suggestions for Authors

·         Congratulations to the authors, the revised manuscript has become more informative for readers

·         The relevance of this study is due to the growing number of patients with fibromyalgia, especially among young people. Studying the regulation of descending pain in fibromyalgia and the relationship with autonomic dysregulation using functional MRI data will help identify additional treatments for this disease.

·         The work is unique and contains more information about this disease. The authors discovered for the first time that in fibromyalgia there is altered coordination of signaling in regions that are involved in pain processing and autonomic regulation, primarily between the locus coeruleus (LC), hypothalamus, thalamus, and periaqueductal gray (PAG). Undoubtedly, these results provide further evidence of how neural signaling is altered in FM.

·         The purpose of the study is concrete and specific.

·         Title of study matches the contains.

·         Despite the small sample size, the authors managed to obtain significant results.

·         Сonclusions and discussion based on the results obtained, as well as on the basis of a comparative analysis with previously obtained results of other authors.